# The Application of Artificial Intelligence and Big Data in the Food Industry

**DOI:** 10.3390/foods12244511

**Published:** 2023-12-18

**Authors:** Haohan Ding, Jiawei Tian, Wei Yu, David I. Wilson, Brent R. Young, Xiaohui Cui, Xing Xin, Zhenyu Wang, Wei Li

**Affiliations:** 1Science Center for Future Foods, Jiangnan University, Wuxi 214122, China; dinghaohan@jiangnan.edu.cn (H.D.); x.xin@jiangnan.edu.cn (X.X.); 2School of Artificial Intelligence and Computer Science, Jiangnan University, Wuxi 214122, China; z1370196418@gmail.com (J.T.); cs_weili@jiangnan.edu.cn (W.L.); 3Department of Chemical & Materials Engineering, University of Auckland, Auckland 1010, New Zealand; w.yu@auckland.ac.nz (W.Y.); b.young@auckland.ac.nz (B.R.Y.); 4Electrical and Electronic Engineering Department, Auckland University of Technology, Auckland 1010, New Zealand; diwilson@aut.ac.nz; 5School of Cyber Science and Engineering, Wuhan University, Wuhan 430072, China; 6Jiaxing Institute of Future Food, Jiaxing 314050, China; zhenyuwang@whu.edu.cn

**Keywords:** artificial intelligence, big data, food industry, emerging technologies, smart sensors

## Abstract

Over the past few decades, the food industry has undergone revolutionary changes due to the impacts of globalization, technological advancements, and ever-evolving consumer demands. Artificial intelligence (AI) and big data have become pivotal in strengthening food safety, production, and marketing. With the continuous evolution of AI technology and big data analytics, the food industry is poised to embrace further changes and developmental opportunities. An increasing number of food enterprises will leverage AI and big data to enhance product quality, meet consumer needs, and propel the industry toward a more intelligent and sustainable future. This review delves into the applications of AI and big data in the food sector, examining their impacts on production, quality, safety, risk management, and consumer insights. Furthermore, the advent of Industry 4.0 applied to the food industry has brought to the fore technologies such as smart agriculture, robotic farming, drones, 3D printing, and digital twins; the food industry also faces challenges in smart production and sustainable development going forward. This review articulates the current state of AI and big data applications in the food industry, analyses the challenges encountered, and discusses viable solutions. Lastly, it outlines the future development trends in the food industry.

## 1. Introduction

Artificial intelligence (AI) is a set of technologies that simulate human intelligence, which allows computers to imitate aspects of human thinking and behavior to achieve autonomous learning, reasoning, planning, and decision-making. The core of artificial intelligence is machine learning (ML), deep learning, natural language processing, computer vision, and other technologies, which can be applied to a variety of fields and industries.

The term ‘big data’ refers to the large and diverse collection of data [1] typically generated by a plethora of sensors or mobile devices, scraped from the internet and other sources, and includes structured and unstructured data, such as text, images, and videos. These data collections are usually characterized by high speed, high density, and high dimensionality, so they need to be stored, processed, and analyzed using bespoke technologies, often now referred to as ‘big data technologies’. In the food industry, big data analytics can help predict market demand, optimize the supply chain, improve food safety and quality, and bring more opportunities and competitive advantages to businesses [2]. Artificial intelligence and big data, as two important technologies that complement each other, play a key role in the food industry and are making a significant impact on innovation and development [3].

Currently, artificial intelligence is widely used in healthcare [4], finance, transportation, manufacturing, and the gaming and processing industries [5,6]. The application of AI in these industries can improve efficiency, reduce costs, and improve service quality.

Artificial intelligence technologies have also been widely used in the food sector. Expert systems, fuzzy logic systems, ANFIS (adaptive neuro-fuzzy inference system) technology, big data, blockchain, and smart sensors are applied to food classification, production development, marketing, supervision, food quality improvement, and supply chain management [7]. The technology has improved food safety and quality [8]. Now, AI technology can also help food companies achieve intelligent production and management, which includes food quality testing, food production control, and food safety monitoring [9].

In addition to artificial intelligence, big data is also an important support that cannot be ignored in the food industry. Big data analytics technology enables companies to extract valuable information from huge data sets and gain insight into key information such as market trends and consumer preferences. Through big data analysis, food companies can better grasp market demand, optimize supply chains, reduce waste, and improve production efficiency. Big data technology can also be combined with other technologies, such as blockchain technology. Blockchain can ensure the traceability of food provenance and quality, transparency from farm to fork, and provide consumers with more reliable product information.

With the continuous progress of technology and the expansion of application scenarios, the application prospects of artificial intelligence and big data in the food industry will be broader. They will become an important means for the transformation and upgrading of the food industry and enhance its competitiveness, bringing more opportunities and challenges to the food industry [10,11].

According to Bradford’s Law [12], selected core journals are regarded as important sources of information and represent the latest scientific and technological advances. In terms of the journal selection of review papers, this review searched the Web of Science database and studied the relevant literature on the application of artificial intelligence (AI) and big data (BD) in the field of food. We systematically refined the retrieved literature with keyword selection (search expression: TS = (food (detection or processing)) and (big data or (artificial intelligence and algorithms)) and year (last 5 years), extracted more than 900 articles, and finally selected more than 150 articles related to artificial intelligence, big data, and food or agriculture to ensure access to the latest research results. Then, by drawing a word cloud (as shown in Figure 1 below), we conducted keyword cluster analysis on relevant articles in the Web of Science database. In Figure 1, high-frequency words are visually highlighted to help readers quickly perceive important information.

The paper is organized as follows. In Section 1, we begin with an introduction to the basic conceptual topics of artificial intelligence and big data, giving an overview of the current state of the food industry. In Section 2, we delve into the application of big data analytics in the food sector, highlighting the potential uses of blockchain technology in terms of security and supply and demand. In Section 3, we discuss the various applications of AI in the food industry and explore its future developments and challenges.

### 1.1. The Early Situation in the Food Industry

In the early stages of the development of the food industry [13], there was a focus on developing more efficient hardware, equipment, and new processing techniques to improve the efficiency of food production and processing. These technological innovations were essential to meet the growing demand for food and to ensure food quality.

During the nascent stages of the food industry, improvements in agricultural tools and techniques were major initiatives to increase the yield and quality of crops. As agricultural production evolved, machinery and equipment such as stone mills and presses were gradually introduced to make processing more efficient and to maintain the freshness and quality of food. At the same time, food processing techniques were improved, and methods such as curing, drying, and smoking were developed to extend the shelf life of food and facilitate storage and transport.

In addition to the development of hardware and processing technology, the effective connection of the entire food chain was also an important concern for the early food industry. The lack of close collaboration and information sharing between raw material producers, processors, and retailers led to inefficiencies in supply chain management, generating excessive inventory and food waste. In order to solve this problem, the food industry has started to establish a more comprehensive supply chain management system [14]. Through the application of information technology, data exchange and information sharing between different segments have been improved, enabling more efficient logistics and inventory management. Meanwhile, through measures such as standardization and quality certification, cooperation and trust between different links in the food industry chain have been enhanced to ensure that the origin and quality of food can be traced.

In summary, the food industry has significantly improved the efficiency of food production and processing from its early stages through the development of more efficient hardware, equipment, and new processing technologies to meet the growing demand for food. At the same time, through the establishment of a more comprehensive supply chain management system, the entire food industry chain has realized effective connection and collaboration, from the production of raw materials to end-users, laying a solid foundation for the development and prosperity of the food industry.

### 1.2. The Current State of the Food Industry

The food industry currently covers a wide range of areas, including food service, food processing, and food retailing [3] and related industries, including agriculture, fisheries, and livestock. It is a large global industry that involves the entire food value chain from production to consumption. Food provides consumers with increasingly diverse choices and demands for safe, secure food. Among them, some new technologies are widely used in the food industry, including artificial intelligence, big data, 3D printing technology, and blockchain technology. The application of these new technologies can improve productivity, reduce waste, improve food quality and safety, increase mass demand, and retain food consumers [15].

### 1.3. The Importance of Food Safety

Food safety issues have always been a focus of attention. Establishing a safety traceability system is key to ensuring food safety and generating revenue for food supply chain components [16,17,18,19]. The food safety cloud [20] is the intersection of food safety work with big data, information technology, and the internet to achieve data and information technology for food quality traceability and identification and to help government departments predict food safety-sensitive information and identify periodic and trending food safety key issues promptly using big data analysis. Meanwhile, it can also provide consumers with information about food’s origin, composition, personalized nutrition programs, standards, and other individualized services and dietary structure advice to help consumers choose safe food and nutritious food, not only to eat safely but also to eat healthily. In the field of food safety [21], machine-learning technology has shown significant improvement in detecting potential food contamination sources. By employing a supervised learning algorithm, we can train the model to analyze multiple variables in the production process, such as temperature, humidity, and chemical composition. This model can identify patterns associated with past food safety incidents and detect anomalies in real-time production. For example, image analysis using convolutional neural networks (CNN) [22] can accurately identify tiny defects or foreign objects on food surfaces, which can help prevent substandard products from entering the market. In the field of food quality [23,24,25], the application of deep learning technology has become the frontier of improving the precision of product quality monitoring. Using sequential models such as recurrent neural networks (RNN) [26] and short-term memory networks (LSTM) [27], multiple parameters can be monitored in real-time on the production line, and fine classification of product appearance and quality characteristics can be achieved. For example, by training convolutional neural networks on high-resolution images, small color changes or shape defects can be detected, thereby improving the overall quality level of the product.

### 1.4. Digital Transformation in the Food Industry

As technology continues to advance, the food industry is undergoing a digital transformation [28]. Automated production and smart manufacturing technologies have improved production efficiency, while big data analytics have provided better insight into consumer needs. In addition, blockchain technology enhances the transparency and traceability of the supply chain, providing security for food safety. At the same time, emerging technologies such as 3D-printed food and virtual reality also open up new possibilities for future food innovation. These digital trends provide a solid foundation for the application of artificial intelligence and big data, and this article will further explore big data, blockchain, and artificial intelligence and their key roles and advantages in the food industry.

## 2. Big Data in the Food Industry

Big data refers to data sets that are huge in scale, of many types, and difficult to process [29,30,31]. It arises from the rapid advancement of computer technology, the Internet of Things, cloud computing, mobile internet, and other technologies. The development of data has driven the progress of science and technology, and the huge amount of data provides new opportunities and challenges for data analysis. Big data generally has 5V characteristics [32]. Volume refers to the scale of big data. The characteristic of big data is that the amount of data is very large, far exceeding the capacity of traditional data processing methods. Velocity is the speed at which data are generated and transmitted. Variety is the variety of types and formats of data. Value is the ability to extract useful information and insights from big data. Veracity is the accuracy and credibility of the data. This kind of data is very difficult to manage and analyze using traditional databases or data processing tools and requires distributed computing and storage technologies. Data technology covers the collection, storage, management, analysis, and visualization of data. However, big data technology differs from small data technology in that big data technology needs to deal with larger data sets, which require more powerful data collection, storage, management, analysis, and visualization capabilities. Through big data technology, people can better understand customer needs, market trends, product performance, and other information to provide better decision support and market insight for enterprises. In the food field, big data, on the other hand, can provide massive food-related information and analysis results to provide decision support for food companies. Applications of big data in the food field include data analysis, food traceability, food nutrition, and health [33].

Big data has an extremely important position in modern society [34]. It can be widely used in business, government, healthcare, transportation, finance, culture and entertainment, and scientific research. For example, Torre–Bastida et al. [35] discussed the application of big data in the field of transportation and mobility, including urban traffic management, intelligent transportation systems, travel mode analysis, and traffic congestion prediction. The analysis of big data [36] can provide unexpected information and insights that can help companies make business decisions, increase productivity, improve product quality, and optimize the customer experience. The application of big data is particularly evident in the food industry [37]. For example, Zuheros et al. [38] used big data to propose a decision model based on artificial intelligence sentiment analysis that analyzes and evaluates restaurant reviews to recommend restaurants to users. Delanoy and Kasztelnik [39] believe that big data analytics can enable factories that produce food to improve food safety and security. With the development of the computer industry, artificial intelligence and big data technology have also developed significantly. When artificial intelligence and big data are combined, synergies can be created. Artificial intelligence can process big data through techniques such as machine learning, deep learning, natural language processing, and computer vision to discover patterns and laws in the data, thus helping people to better understand the data and grasp the information. These intelligent technologies have had a profound impact on people’s lives and brought a lot of convenience. Especially in the food industry [40], these two technologies have achieved remarkable results in food flavor matching, food safety, and food testing [41,42,43].

### 2.1. Applications of Big Data in the Food Industry

With progress and economic development, people’s requirements for food have become more stringent. People are now more concerned about food flavor and food nutrition [44]. Traditional food science and technology fail to meet the specific food needs of each region and each individual. The emerging artificial intelligence and big data analytics technologies have brought revolutionary changes to the food industry in terms of supply chain optimization and food safety. The internet, combined with artificial intelligence and big data, is the future direction of food safety, and big data is an important factor in food safety [45]. For example, Kazama and Sugimoto [46] have used big data technology to form a neural network that converts food from one country’s recipes to another’s recipes based on its ingredients and composition, completing the transformation of regional food styles and making it easier for people to taste cuisines from exotic places. Kalra et al. [47] built a big data-based nutritional assessment system that allows people to give higher priority to the nutritional content of the food on their plates and calculate the nutrition of food through recipe analysis.

In addition, Whitehouse et al. [48] used big data technology (surveillance cameras) to monitor kitchen operations, improving food safety and providing greater peace of mind for guests sitting at the table. Not only from the food point of view, AI and big data technologies can also advise diners to choose quality restaurants and help restaurant managers make the best and most rational decisions. Lee et al. [49] developed a predictive model of restaurant reviews based on big data that is useful for customer decision-making and to help restaurant managers make the best and most rational decisions. The integration of AI and big data brings benefits to small-scale food producers or localized food systems, as described by Ajit Maru in 2018 [50], using AI and big data to design tools and applications tailored to their specific circumstances and capabilities to make data-driven agriculture [51] friendlier to them. Through real-time monitoring and feedback, local food producers can adjust their production strategies more flexibly to better meet local market needs. In addition, by establishing a blockchain-based traceability system, small-scale producers can provide more transparent and traceable product information, increasing consumer trust in local products, thereby promoting the sustainable development of local food systems. Figure 2 illustrates the different applications of big data in the food sector, including food safety and traceability, consumer insight and market analysis, product optimization and quality control, and innovation and sustainability. Figure 3 also illustrates five aspects of big data security in the food industry and the processing model of big data.

#### 2.1.1. Application of Personalized Marketing and Recommendation System

Personalized marketing is a customized marketing strategy and recommended content based on consumers’ interests, preferences, and behavioral characteristics. For example, Liao et al. [52] explored how marketing campaigns in social networks can be analyzed with big data to understand user behavior, preferences, and interaction patterns. The application of personalized marketing and a recommendation system can be realized through big data analysis and artificial intelligence algorithms.

With the help of big data analytics, companies can understand consumers’ purchase history, preferences, and behaviors to provide personalized product recommendations, pricing strategies, and promotions that enhance consumers’ buying experience and loyalty [53].

A recommendation system [54] uses algorithms and models to recommend food products that suit consumers’ tastes and needs based on their preferences and similar consumer behaviors, increasing purchase conversion rates and sales.

#### 2.1.2. Consumer Behavior Analysis and Forecasting

Big data analytics can help companies better understand consumer behavior patterns, decision-making processes, and purchase motivations and, thus, predict trends in consumer behavior and changes in preferences [55]. Through consumer behavior analysis, companies can identify the characteristics and needs of different consumer groups and optimize product positioning, marketing, and channel strategies. Predicting consumer behavior can help companies make more accurate inventory management, production planning, and supply chain decisions and improve operational efficiency and the ability to meet consumer demand. Through the application of personalized marketing and recommendation systems and consumer behavior analysis and prediction, food companies can better understand consumers, provide personalized products and services, enhance consumer satisfaction and loyalty, and, thus, gain an advantage in the market competition.

#### 2.1.3. The Utilization of Big Data Analytics in Supply Chain Management

Big data analytics holds a significant position in food supply chain management [56]. By collecting and analyzing large amounts of supply chain data, companies can gain insights into inventory, procurement, production, distribution, and more. Big data analytics can help companies optimize supply chain processes, improve operational efficiency, reduce inventory costs, and decrease transportation time. With the technology and tools of big data analytics, enterprises [57] can conduct real-time monitoring and forecasting of the supply chain, identify potential risks and bottlenecks, and take appropriate measures to adjust and optimize. The application of AI and big data in the food supply chain has also proposed specific solutions to supply chain inefficiencies and food waste. As Joshi pointed out in 2020, through advanced predictive analysis of machine learning [58], accurate prediction of demand fluctuations can be achieved, and then real-time adjustment of optimal inventory levels can be achieved through intelligent inventory management systems. In addition, big data analysis plays a role in optimizing production plans [59], integrating information such as market trends, raw material supply, and production capacity to ensure the precise matching of production and market demand. Real-time monitoring and feedback mechanisms improve the visibility and responsiveness of the supply chain through IoT (Internet of Things) sensors and advanced data analytics, helping to identify and resolve problems in a timely manner. These advanced applications provide strong support for improving supply chain efficiency and reducing food waste.

#### 2.1.4. Application of Forecasting Models and Machine Learning Algorithms in Demand Forecasting

Demand forecasting [60] is a critical component of decision-making in the food industry. By forecasting changes and trends in consumer demand, companies can plan production and supply rationally and avoid overstock or out-of-stock situations. Forecasting models and machine learning algorithms can use historical sales data, market trends, and other relevant factors to build accurate demand forecasting models. By analyzing and learning from the data, these models and algorithms can automatically identify and capture demand patterns, make forecasts, and provide targeted recommendations and decision support.

Through the application of big data analytics in supply chain management and predictive models and machine learning algorithms in demand forecasting, food companies can more accurately understand the supply chain situation and consumer demand, achieve matching of supply and demand, and improve operational efficiency and customer satisfaction. This will help optimize supply chain management [61] and production planning and enhance the competitiveness of companies.

### 2.2. The Bottleneck of Big Data Applications for the Food Industry

Big data technology is now extensively employed in the food sector [62], including many aspects of production, sales, and consumption. Although big data technology can bring many advantages, such as increasing production efficiency, improving marketing strategies [63], and enhancing product quality, there are also some disadvantages. These include data privacy issues, quality control problems, and bias problems. In the process of big data analysis, a great deal of data must be collected and analyzed, including consumers’ personal information, behaviors, and preferences. These data may be illegally accessed and misused, posing a risk of privacy leakage to consumers. Although big data analytics can help producers better understand consumer preferences and needs, it may sometimes lead to producers overemphasizing market demand at the expense of product quality and safety, thus posing a health risk to consumers. Big data analytics is based on historical data and is used to make predictions and analyses, so there may be some bias. If producers make decisions based on data analysis alone, without considering the impact of other factors, the final decision may be wrong or risky.

Although big data technology can facilitate the development and progress of the food industry, it must be applied carefully to avoid negative effects such as data privacy issues [64] and quality control problems. Meanwhile, the government should strengthen regulation and legislation to protect the legitimate rights of consumers.

Blockchain technology is considered a viable solution to address the challenges and implications of big data [65]. ‘Blockchain combined with Big Data’ is considered to be an effective solution for sharing data. The changes brought by blockchain technology on the food supply chain include food security, food safety, food integrity, smallholder support, monitoring and management, etc. It also employs technologies for verifying and digitally signing documents, verifies and tracks ownership of intellectual property and proprietary systems, enables smart contracts [66], and tracks patient health records. Decentralized ledger solutions leveraging blockchain technology can also be linked with smart contracts and decentralized applications [67]. As De Filippi explained in 2016 [68], the decentralized nature of blockchain ensures distributed storage of data, eliminating a single point of attack and, thus, reducing the risk of data being hacked or tampered with. Each block contains information from the previous block, forming an immutable chain structure that provides a reliable guarantee of data transparency and integrity. Secondly, the blockchain’s smart contract capabilities can be used to formulate and enforce data access rights [69]. With smart contracts, it is possible to specify which parties have the right to access, modify, or share specific types of data. This method ensures that data access can be traced and controlled and improves the management level of data privacy.

### 2.3. Blockchain Technology

Blockchain technology is one of the emerging technologies currently receiving a lot of attention, and it brings many benefits to the agri-food supply chain [70]. Blockchain, derived from Bitcoin, is a decentralized database recorded in the form of cryptographic blocks for the execution and sharing of every transaction or digital occurrence in a public ledger, whose information can be authenticated at any time in the future. The core features of blockchain are data immutability and decentralization, which makes the information recorded more authentic and reliable and effectively solves the problem of mutual distrust. In the field of food traceability, blockchain can achieve traceable supply chain management by assigning a unique digital identifier to each food product [71,72]. This includes the growing conditions, batch number, and expiration date of the food. This will not only help reduce food waste but also help consumers assess the ecological footprint of their food and manage the redistribution of extra food. In addition, food authentication in the food supply chain [73] (such as food ingredients, production date, production source, processing, and technology used) has been a focus of attention, and traditional authentication methods often have problems such as fraud, while the use of blockchain technology can effectively avoid such problems [71,74].

Lin et al. [75] proposed a framework for building a food traceability system with smart contracts built upon blockchain technology and the Ethereum platform. The framework is explicitly applied to the food industry, using blockchain technology to ensure transparency and safety in food production and supply chains. Mohan et al. [76] proposed a model that combines existing food quality systems and technologies at all stages of the supply chain and uses blockchain technology to propose a solution for food (chicken) tracking. Kim and Laskowski [73] used a blockchain modeling approach that integrates IoT devices for capturing and sharing data from supply chain sources. The study also addresses the issue of the existence of common data standards at different stages of the supply chain when using decentralized blockchain networks. On the other hand, Kumar and Iyengar [13], built a rice supply chain system built upon blockchain technology that aims to improve security in the management process. This case is directly relevant to the food industry, using blockchain technology to solve security issues in the rice supply chain. However, it is also important to recognize that not all blockchain applications are equally focused on food safety issues, while some may focus on other aspects such as finance and logistics.

Blockchain technology, with its reliable, secure, distributed-based nature, provides a monitored and managed full-chain solution in the agri-food supply chain from the production of raw materials to the store shelves, involving producers, consumers, suppliers, and regulators, and greatly improving food traceability [77,78]. In addition, with smart contract technology, manufacturers can further reduce costs and enhance the overall efficiency of the manufacturing industry. It is also important to mention that the potential transparency offered by blockchain technology can also help facilitate the development of a reputation-based trading system. In the context of blockchain systems applied to food supply chains, the communication and processes are typically represented as one-way. Figure 4 illustrates the process of the food supply chain in a blockchain system and the behavior of different participants, and Table 1 illustrates how blockchain can guarantee transparency and traceability in the food supply chain, improve food safety and product quality, and enhance consumer trust in products.

However, blockchain technology also faces future challenges and issues to be addressed [84]. With the integration of an increasing number of components into blockchain systems, such as RFID, smart sensors, smart robots, biometric data, IoT, and big data, the underlying logic and implementation may become more complex. In addition, although blockchain technology is already widely applied in food supply chains around the world, operation and maintenance costs are increasing, which may limit the entry of new suppliers into the market. In addition, large companies may adopt privately licensed blockchain technology, which may lead to giants and oligopolies monopolizing the market. In addition, how to improve the regulatory system related to blockchain technology is still an issue that needs to be further debated. Currently, there is no consensus among top policymakers and experts on how to properly apply blockchain technology and cryptocurrency transactions.

## 3. Artificial Intelligence in the Food Sector

When we talk about artificial intelligence in food, we are in a field full of cutting-edge technology and limitless potential. With the rapid development of science and technology, artificial intelligence has profoundly affected all levels of the food industry. From expert systems to fuzzy logic systems, from ANFIS technology to NIRS technology to CVS technology and the application of artificial intelligence and sensors in the food industry, these innovations are bringing unprecedented changes to food production, quality control, and safety [85,86]. Compared with big data, these technologies have unique characteristics and application scenarios, and each plays a unique role in solving problems and optimizing processes.

### 3.1. Knowledge-Based Expert Systems in the Food Industry

In the food industry, expert systems, as a knowledge-based artificial intelligence technology [87,88], can simulate the thinking style and knowledge structure of experts to achieve automated decision-making and problem-solving [89]. The system can use the knowledge and experience of domain experts, combined with techniques such as machine learning and natural language processing, to build a system with certain reasoning capabilities to automate the solution of complex problems. For example, expert systems can be used in applications such as assessing food quality, detecting food safety problems, and optimizing production processes.

In addition, some expert systems can continuously learn and improve [90]. As data and knowledge accumulate, the system can update and optimize itself, thereby improving its problem-solving accuracy and efficiency. A knowledge-based system, alternatively referred to as an expert system, is a computer application that integrates a large number of problem solutions associated with a particular domain. The system is capable of mimicking the decision-making process of human experts. Typically, it comprises six components: the human–computer interface, knowledge base, inference engine, interpreter, comprehensive database, and knowledge acquisition module, among which the most important are the knowledge base and the reasoning machine. The knowledge base stores a large number of facts, objects, cases, and rule conditions, which are expressed in the form of “IF-THEN” statements. The reasoning machine can perform specific operations on this knowledge and generate solutions to various problems with the help of human experts.

Since food safety is a top priority in the food production industry, expert systems have been applied in the production of food, quality testing, and food risk assessment [91]. In addition, expert systems are also used in the food industry, such as web-based expert systems for diagnosing pests and diseases of banana plants [92], intelligent expert system databases [93] for automatic control of product quality indexes, and expert systems based on fuzzy logic models [94] applied to the coffee industry.

#### The Future and Challenges of Expert Systems

In the coming years, we predict knowledge-based expert systems will be combined with techniques such as fuzzy logic and neural networks for advanced control processes in food processing, control modeling, and multivariate and nonlinear processes. In particular, the ability of fuzzy logic to handle uncertainty in hybrid fuzzy expert systems may bring significant advantages. In the food field, the application of knowledge-based expert systems has the potential to help companies reduce production costs, improve productivity, optimize product quality, and enhance market competitiveness. In the 1990s, due to the lack of hardware and software conditions, expert systems did not reach their full potential. Today, with the rapid development of hardware and software and the progress of artificial intelligence and machine learning, expert systems are highly practical and have been applied to engineering, medicine [95], business, food, and other fields; it is an integral part of artificial intelligence [96].

### 3.2. Fuzzy Logic Systems

Traditional knowledge-based general expert systems have a distinct disadvantage in that they are unable to handle problems that are beyond the scope of their knowledge database. When the system is confronted with problems that are not included in the knowledge base, the rule-based system will not be able to provide deterministic results. To solve the problems of traditional expert systems in dealing with uncertain and fuzzy data, Zadeh first introduced the concept of fuzzy logic in 1965. A fuzzy logic system is an artificial intelligence technique based on the principles of fuzzy logic [97,98,99]; its main use is to deal with problems involving fuzzy concepts. Unlike traditional logic systems, fuzzy logic systems allow the truth value of propositions to take any value between 0 and 1, thus allowing better handling of problems with fuzzy boundaries. In addition, fuzzy set theory provides a convenient means of dealing with uncertainty and transforming expert knowledge into computer-processable quantitative functions. Traditional analytical and statistical methods often struggle to deal with expert knowledge, but complex mathematical relationships are not required in building fuzzy logic applications. Fuzzy models can be expressed in easily comprehensible linguistic rules, resembling the format of rule-based expert systems. The emergence of fuzzy logic systems introduces the stereotypical judgment and imprecise nature of human judgment and can improve the generalizability of expert systems.

Fuzzy logic translates variable values into a linguistic representation, where the interpretation corresponds to a fuzzy set, and based on these representations, the fuzzy system then proceeds to the next step of judgment. Fuzzification [100] is a process in which explicit values are converted into affiliations, and a fuzzy input set is generated. The correspondence of a membership function fuzzy system is usually between 0 and 1. Fuzzy rules are also called “IF-THEN” rules [101], where IF is the necessary precondition and THEN is the corresponding result. The fuzzy logic system consists of the following steps: first, the actual values are transformed into fuzzy values, and then rules are applied to map these fuzzy inputs to fuzzy outputs. Next, a fuzzy output is obtained by combining the results of the fuzzy rules through statistical methods. Finally, the fuzzy output is transformed into specific actual values through the process of defuzzification. These steps work in concert to form the basic operational flow of a fuzzy logic system. Figure 5 illustrates these steps in the operations of a fuzzy logic controller.

Fuzzy logic systems are efficient and simple designs for fast analysis and problem-solving with a highly accurate approach. As a result, these systems have been widely used in industry [94,100,102,103]. In the food industry, fuzzy logic techniques are also applied. For example, Farzaneh et al. [104] proposed an adaptive neuro-fuzzy inference system and applied it to canola oil extraction modeling. Samodro et al. [105] used fuzzy logic techniques to help a coffee roaster maintain the smell and quality of coffee. Yulianto et al. [106] implemented a fuzzy inference system and used it for salt yield estimation. Basak et al. [107] employed fuzzy logic techniques to assess the concentration of essential oil in withered leaves and its efficacy as a preservative for fruit juices. Vivek and Subbarao [108] utilized fuzzy logic techniques in the sensory evaluation of food items. Furthermore, emerging approaches such as fuzzy set theory have proven effective in evaluating the sensory attributes of diverse traditional and innovative foods developed [109] through fortification and modified processing methods.

In fuzzy modeling, linguistic entities such as “unsatisfactory, fair, moderate, good, and very good” are used to describe sensory attributes (including color, aroma, taste, texture, and mouthfeel) of food products obtained through subjective evaluation. Shahbazi et al. [110] proposed a food traceability system based on blockchain machine learning that combines a fuzzy logic traceability system based on shelf life management systems to manipulate perishable foods to address issues regarding food resolution such as lightweight, evaporation, warehouse transactions, or shipment time. With the application of fuzzy logic systems, problems with fuzzy boundaries can be better handled, leading to more accurate and objective decisions.

### 3.3. Adaptive Neuro-Fuzzy Inference System (ANFIS) Technology

The adaptive neuro-fuzzy inference system [111] is an innovative structure for fuzzy inference systems that seamlessly integrates fuzzy logic and neural networks. It is a technique that adjusts the premise parameters and conclusion parameters using a hybrid algorithm of least squares and backpropagation and can automatically generate IF-THEN rules. It combines the computational learning capability of ANN (artificial neural network) networks with the human-like reasoning capability of a fuzzy logic system. Where ANFIS comes into play is reflected in the fact that, in some cases, it can automatically determine the appropriate parameters of the membership functions and does not need to set the appropriate affiliation functions and their parameters by itself. This is especially the case when we already have a set of inputs and associated output variables and values. Like artificial neural networks (ANN), ANFIS systems can “automatically” adjust their nodes and the connections between them. The five-layer structure of ANFIS consists of a fuzzy layer, a product layer, a normalization layer, a defuzzification layer, and a total output layer. Figure 6 illustrates the network structure of ANFIS.

In Figure 5, *f*_i_ is the output result corresponding to rule i; A1 and B1 are the nonlinear parameters corresponding to rule i; Wi represents the trigger ratio of rule i in all rule bases.

The functions of the nodes in each layer are as follows.

Layer 1: The fuzzification layer, where input variables are transformed into fuzzy sets, and the output represents the degree of membership in the fuzzy set. A1, A2, B1, B2 belong to fuzzy set A. It is responsible for fuzzifying the input features x and y using membership functions to obtain a degree of membership in the range of [0, 1].

Layer 2: The rule activation layer, which implements the operation of the fuzzy set in the antecedent part of each rule. Each node in this layer is fixed and computes the strength of each rule by multiplying the degree of membership of each feature. The output of each node represents the activation strength of a rule, and the node function can be implemented as taking bounded products or strong products.

Layer 3: The normalization layer, which normalizes the activation strengths of each rule to indicate their relative weights in the entire rule base.

Layer 4: The defuzzification layer, the result of the calculation rule, is generally given by a linear combination of input features.

Layer 5: The output layer calculates the exact output by defuzzifying the aggregated outputs of the rules. The single node in this layer computes the sum of all incoming signals as the final output of the system.

With the progress made in fuzzy logic and neural networks, the adaptive neural network fuzzy inference system, as a product combining these two theories, has emerged as a significant research direction in the field of computational intelligence due to the combined benefits of the expressive nature of fuzzy logic and the self-learning capability of neural networks. In the field of food testing, this technique has had an impact [107]. For example, researchers have used the ANFIS model to study the sensory attributes of ice cream [108] and predicted the acceptability of the taste based on the input parameters. The model achieved a minimum error rate of 5.11% and a correlation coefficient of 0.93. In a separate study, another researcher employed the ANFIS model to forecast the quality of virgin olive oil samples [112]. These results show that combining fuzzy inference systems with neural networks, especially in food testing, has great advantages. In addition, Abbaspour-Gilandeh et al. [113] designed a system combining artificial neural networks and ANFIS for predicting the kinetic energy and energy of quince under a hot air dryer, Kaveh [114] designed an ANFIS model for predicting the moisture diffusivity and specific energy consumption of drying potatoes, garlic, and melons under convection hot air dryers, Arabameri et al. [112] utilized an adaptive neuro-fuzzy inference system (ANFIS) to assess and forecast the oxidative stability of virgin olive oil. Mokarram et al. [115] employed an adaptive neuro-fuzzy inference system and multivariate linear regression to estimate the flavor of oranges. Table 2 lists the published ANFIS technology applications in the food industry.

### 3.4. Near-Infrared Spectroscopy Technology Combined with Artificial Intelligence

Near-infrared spectroscopy (NIR) is a non-invasive analytical technique based on optical principles that can be used for rapid and accurate analysis and detection of the chemical and physical properties of substances, which uses the absorption and scattering properties of the substance molecules in the near-infrared spectral region to analyze the chemical composition, structure, and properties of substances by detecting the intensity of the absorbed and scattered light. NIR technology can be used for rapid and accurate analysis of food composition and quality indicators [118]. Compared with previous techniques, NIR technology has the advantages of no chemical substances, fast and accurate results, non-destructive, low cost, and resource-saving, thus becoming a viable alternative to traditional techniques. In the past few years, the fusion of artificial intelligence algorithms and NIR technology has achieved effective monitoring of food internal quality detection and disease conditions, mainly using methods such as least squares and multiple linear regression. In the food field, NIR technology has been widely applied in food quality inspection and quality control. For example, the NIRS system combined with artificial intelligence technology can classify and detect food products, and NIR technology can detect mechanical damage to mangoes [119]. In 2020, Curto et al. [120] used artificial neural network-based near-infrared spectroscopy to precisely forecast the sensory characteristics of cheese; this technology provides an efficient and reliable quality assessment tool, but its application is not limited to cheese, it is also applicable to the quality control of other foods. Gunaratne et al. [121] used NIR spectroscopy and machine learning modeling for chocolate quality assessment; it provides a scientific basis for improving the quality and consistency of chocolate products. At the same time, this technology also provides a reference for other food composition analyses and quality assessments. Qiao et al. [122] used a hyperspectral imaging system to assess pork quality and marbling levels; it provides a new perspective on food quality control. This technology not only has a wide application in meat quality but also can be used for quality monitoring of other agricultural products, bringing new possibilities to the agricultural field. Alshejari et al. [123] designed an intelligent system for decision support in identifying meat spoilage through the analysis of multispectral images; it also provides an innovative solution to other food freshness and preservation problems.

In conclusion, NIR technology is fast, accurate, non-invasive, and low-cost, which can enhance the quality and market competitiveness of food products. With the continuous development and popularization of artificial intelligence technology, the application prospects of NIRS technology in the food field will be broader.

### 3.5. Application of Computer Vision Systems in the Food Industry

Computer vision is an artificial intelligence technology that utilizes image processing and pattern recognition techniques to automate the analysis and interpretation of visual data, such as images and videos. It aims to simulate human vision by enabling computers and related devices to understand digital images and videos, extract meaningful information, and make informed decisions. It involves the development of sophisticated algorithms, including traditional methods and deep learning approaches, to enable computers to perceive, analyze, and comprehend visual data in a manner similar to humans. The underlying principle of computer vision is to enable a computer to process and comprehend images at the pixel level, utilizing specialized software algorithms to retrieve, process, and interpret visual information effectively. Its main task is to acquire images and videos and process them with a computer to obtain three-dimensional information about the corresponding scene, including image segmentation, image classification, and image detection. The earliest experiments with computer vision began in the 1950s when it was used to parse typed and handwritten text. At that time, the analysis procedure was relatively simple and required a lot of manual operations, where the operator had to provide data samples manually for analysis. The traditional manual operation made it difficult to provide large amounts of data, and coupled with limited computing power, the error rate of such analysis was quite high. Today, we do not lack powerful computing power. Cloud computing, coupled with powerful algorithms, can help us solve even the most complex problems. However, it is not only the combination of new hardware and advanced algorithms that are driving the development of computer vision technology; it is also the vast amount of publicly available visual data that we generate every day that is driving the technology. Computer vision has a wide range of applications, including but not limited to medical image detection, autonomous driving, facial recognition, agriculture, and food inspection. In the food field, computer vision technology has been widely used [124]. Computer vision can be used in food production lines for automated inspection and quality control and can also be applied to food quality inspection. Lopes et al. [125] developed a computer vision classification system for barley flour that incorporates spatial pyramid segmentation integration. Phate et al. [126] employed a computer vision system to cluster the ANFIS weighing model for sweet orange (*Citrus limetta*); Nadim et al. [127] applied image processing techniques to quality control in mushrooms; Villaseñor-Aguilar et al. [128] used an artificial vision system for sweet pepper for ripeness assessment for quality control. Among the various deep learning neural networks, convolutional neural networks have been applied to computer vision, and it is particularly suitable for handling various tasks in the field of computer vision, such as image classification, object detection, and semantic segmentation. With advances in machine learning, especially the development of convolutional neural networks, collected food pictures can be further processed to classify food. Figure 7 illustrates the use of a convolutional neural network (CNN) in computer vision to automatically extract food shape features for food recognition. Table 3 lists published computer vision applications in the food industry.

### 3.6. Artificial Intelligence Combined with Smart Sensors for Real-Time Inspection in the Food Industry

In today’s food industry, guaranteeing product quality and safety holds utmost significance. With the ongoing advancements in artificial intelligence and smart sensor technology, it is now feasible to monitor and assess the quality and safety of food products in real-time. Artificial intelligence analyzes large amounts of data through machine learning algorithms to predict possible problems and identify and solve food safety issues promptly. Meanwhile, smart sensors can monitor parameters such as temperature, humidity, pressure, and odor in real-time to detect potential problems. The application of these technologies enables the food industry to more effectively ensure product quality and safety, bringing better health and quality of life to consumers.

The integration of artificial intelligence and smart sensors [137] allows for real-time monitoring and early warning to ensure food safety. Once the sensor detects an abnormality, the system immediately notifies the operator and takes appropriate action to avoid food safety problems. In addition, the sensor data can be analyzed to better understand the characteristics of product quality and process changes so that the production process and product quality can be continuously optimized. The effectiveness and practicality of this technology application open up a wider scope of development for the contemporary food industry and provide a reliable guarantee for safeguarding public health.

Smart sensors are the core foundation of artificial intelligence technologies and can be divided into different types, such as physical sensors, chemical sensors, and biological sensors. Artificial intelligence technologies that can enhance sensor systems encompass a variety of methodologies, such as knowledge-based systems, fuzzy logic, automatic knowledge acquisition, neural networks, genetic algorithms, case-based reasoning, and ambient intelligence. These intelligent techniques contribute to the capabilities of sensor systems, enabling them to acquire, process, and interpret data more effectively. These artificial intelligence technologies can be widely used in scenarios such as small sensor systems and single sensors.

The appropriate utilization of artificial intelligence technology can enhance the competitiveness of sensor systems and applications. One common approach involves integrating sensors for real-time detection, such as electronic nose (E-nose), electronic tongue (E-tongue) [138], and machine learning. These sensor integrations are often combined with technologies such as artificial neural networks to enable advanced data analysis, pattern recognition, and decision-making capabilities. This integration enables the development of intelligent sensor systems that can effectively analyze complex data and make accurate real-time assessments in various domains [138]. The first of these technologies is used to monitor food products in real-time, achieve high accuracy, and issue alerts in a short time so that timely action can be taken to safeguard food safety and quality [139]. The application of these technologies can not only improve the efficiency and productivity of the food industry but also provide the public with safer and healthier food.

In recent years, there has been significant progress in the implementation of smart sensors incorporating artificial intelligence techniques in the food sector. For example, McVey et al. [140] developed a spectroscopy-based smart sensor. Ndisya et al. [141] used an optical sensor technology based on hyperspectral imaging to assess the quality variation of purple-speckled yam slices in dry hot air and successfully developed a predictive model. This research is a clear example of applications in the food industry, but the same hyperspectral imaging technology can also be used in agriculture to monitor plant growth and predict disease. In addition, Abedi-Firoozzah et al. [142] applied kale anthocyanins in biosensors and food packaging. Sanaeifar et al. [143] used an electronic nose to classify, detect, and identify different types of fruits as well as to detect defective parts in food products. Feng et al. [144] designed an electronic nose that can be used to detect spoilage of food products such as meat and fish. The electronic tongue is an intelligent sensor capable of analyzing and identifying food through the sense of taste. It can identify different chemical substances, both organic and inorganic, through eight built-in electronic sensors. Electronic tongue technology is used in a wide range of applications in the food sector, including food traceability, food freshness, food quality grading, and quality monitoring during food production. For example, Wadehra and Pati [145] detail the application of electronic tongues in different food production and processing processes, such as fruit juices, dairy products, vegetables, and fruits. The electronic tongue technology is also capable of quickly distinguishing between chicken breeds and product quality and will, therefore, become an important tool for the evaluation of chicken meat and its product quality. An electronic tongue system developed in China uses bare metal electrodes such as gold, silver, and platinum as working electrodes and can be used in combination with suitable data analysis methods for quality differentiation of wine, liquor, tea, meat, and cow milk, as well as qualitative and quantitative analysis of food-borne pathogenic bacteria and pesticides. Through sensor technology and the Internet of Things, key parameters and indicators in the food production process can be automatically collected and recorded, reducing errors from manual intervention and providing a more accurate and comprehensive database. These data can be used for production process optimization, quality control, and food safety, helping companies achieve intelligent production and refined management. Table 4 shows a comparison of conventional laboratory instruments, electronic nose, electronic tongue, computer vision, and sensory analysis. Where ‘√’ represents meeting this condition, and ‘×’ represents not meeting it.

## 4. Future Trends and Challenges for Artificial Intelligence Applications in the Food Field

In recent years, the food industry has undergone rapid and dramatic changes thanks to the food industry revolution [146]. With the development of artificial intelligence, the future of the food industry will be based on technologies such as smart agriculture, robotic agriculture, drones, 3D printing, and digital twins [147,148,149,150]. Meanwhile, thanks to robotics and automation in the sustainable food industry [151], the food manufacturing industry is shifting from traditional manual production to an automated production phase. Within this change, packaging, warehousing, distribution, marketing, and consumer service are also moving toward automation. However, the emergence of these new models places higher demands on the workforce of the food industry. In addition, food safety issues have become a major international concern [152,153]. The development of IoT technology [154,155] can help solve this problem as it can identify products and trace them back to every step of food production and processing. The concept of Industry 4.0 for food processing represents an improvement in the quality and safety of processed foods in the current digital age by leveraging the Fourth Industrial Revolution (called Industry 4.0) technologies. Industry 4.0 for food processing technology has gained a lot of attention in recent years, revolutionizing and transforming many aspects of the food industry, including food processing and food inspection. The introduction of Industry 4.0 for food processing to the food industry has led to an increase in the utilization of artificial intelligence in the food sector, which addresses the various issues that arise in the food sector through innovative approaches. Industry 4.0 for food processing [156,157] integrating technology or intelligent systems into traditional industries enhances the safety and quality of processed food, standardizes production processes, reduces production costs and time, conserves energy and resources, and minimizes food loss and waste. Figure 8 introduces the elements related to AI and big data in Industry 4.0 for food processing.

More and more researchers are now conducting in-depth research in the food sector, more AI methods will emerge, and the amount of AI applications will continue to rise in a straight line. Here are some of the latest and most cutting-edge AI technologies being applied to the food industry, covering synthetic food, food production and energy efficiency management, supply chain management, sales forecasting, assisted cooking, and personalized nutrition. Compared to traditional naturally derived ingredients, machine learning and biotechnology can be used to create new synthetic foods using technologies similar to 3D printing, such as meat-free plant-based meat and egg-free cakes. By using the Internet of Things, deep learning and computational vision technologies can minimize waste energy and material resources, making production processes more efficient and environmentally friendly. At the same time, AI can also enhance the quality of food production by automatically controlling details of the production process, such as temperature, humidity, and ventilation, based on monitored environmental parameters. AI technologies can intelligently track large supply chains to ensure food traceability and improve corporate response to crises by monitoring risk factors such as weather and natural disasters that may affect food production and distribution, which is called supply chain management. Sales forecasting refers to feeding large amounts of data into AI models (such as the recently popular ChatGPT 4.0 model [158,159]), which can accurately predict consumer buying behavior. For example, factors such as food recipes, tastes, and weather drive consumer demand based on consumer buying habits and calendar events to promote sales to achieve personalized sales forecasts. The use of computer vision technology can help chefs design dishes and improve ingredient recognition and the quality of dishes, which is called assisted cooking. Nowadays, with the widespread popularity of smart bracelets, weight scales, and sports watches, users’ physical data can be accessed by these devices, and they can also be combined with artificial intelligence technology to analyze users’ physical conditions and provide more personalized health management solutions.

### 4.1. Future Development Direction and Outlook

#### 4.1.1. Application Prospects of Emerging Technologies

Applications for emerging technologies include augmented reality, virtual reality, blockchain, and edge computing. Augmented reality (AR) and virtual reality (VR) technologies can provide immersive food experiences that help consumers better understand the origin, preparation process, and nutritional information of products, enhancing consumer engagement and trust. Blockchain technology can provide food traceability and trustworthiness, ensuring transparency and security in the food supply chain. Consumers can track information on the origin, processing, and transportation of food to ensure the quality and traceability of products. Edge computing pushes data processing and analysis to edge devices close to the data source, allowing real-time data processing and decision-making at places such as food production sites or farms, improving production efficiency and quality control.

#### 4.1.2. Possible Directions for Innovation and Improvement

Possible directions for innovation and improvement cover the application of deep learning and reinforcement learning, as well as multimodal data fusion. By further developing deep learning and reinforcement learning algorithms, the prediction and decision-making capabilities of AI systems in the food industry can be improved to better address production, quality control, and risk management issues. Multimodal data fusion technology will integrate multiple data sources (such as images, text, sensor data, etc.) to provide more comprehensive food information and analysis results to help decision-makers better understand and manage food production and supply chains.

#### 4.1.3. Exploration of Feasibility and Sustainability Issues

With the increasing amount of data in the food industry, it has become particularly important to protect the privacy and security of data for consumers and businesses. Appropriate data protection policies and technical measures are needed to guarantee the secure storage and transmission of data. In addition, training professionals with AI and big data analysis skills are key, and there is also a need to popularize and promote related technologies so that more businesses and farmers can benefit from AI and big data applications. As we advance AI and big data applications, it is imperative to intensify our focus on mitigating their adverse environmental impacts. Proactively exploring eco-friendly data center solutions and optimizing energy consumption strategies becomes paramount to ensure a sustainable trajectory of our progress within the natural world. These directions and issues provide an outlook, but the sustainability of actual developments and solutions requires further research and practice.

#### 4.1.4. Future Challenges Ahead

In the future, the food industry will face numerous challenges. The production process generates a vast amount of data. Industry 4.0 for food processing plays a role in capturing these data, and big data plays a crucial role in collecting and processing these data, which poses a challenge for pre-processing. AI technology is used to evaluate and analyze the data produced by the food industry, but the data processed by AI often need to be manually labeled. As for blockchain, it is a relatively new and advanced technology that may encounter technical problems, cost issues, and data security and authenticity issues. AI in the food industry also faces challenges such as data privacy and security, technical complexity, and integration with traditional methods. In terms of data privacy and security, new technologies may face potential data breach risks when processing large amounts of consumer information [160]. In addition, due to the technical complexity, the maintenance and updating of AI systems may require highly specialized skills, which may limit its adoption by small businesses or farmers. When integrated with traditional methods, production processes may be disrupted or uncoordinated due to technical differences. When adapting to actual production conditions, AI systems may be affected by environmental changes. For example, certain technologies may suffer from reduced performance in extreme weather conditions, resulting in reduced productivity. In addition, if the algorithm relies on large-scale data sets that fail to adequately account for local factors, applications in specific regions may produce inaccurate results. Therefore, we need to pay more attention to these potential problems and take appropriate measures to address or mitigate the impact of these threats and vulnerabilities. The implementation of artificial intelligence (AI) and big data technologies in the food industry faces a number of limitations and obstacles, especially in terms of technical complexity and integration with existing systems. First, technical complexity, which involves the understanding and application of advanced algorithms, models, and data processing techniques, requires highly specialized technical teams. In addition, as Anagnostopoulos pointed out in 2016 [161], data in the food industry are often fragmented and heterogeneous, and consistency and standardization in processing these data is also a challenge. In this process, ensuring the quality and integrity of these data is particularly important. Integration with existing systems is another prominent issue [162]. Many food businesses already have large information management systems with traditional management approaches, and integrating new AI and big data technologies into these systems requires careful planning and effective change management. Differences in technology architecture, inconsistent data formats, and acceptance of new technologies are also challenges in the integration process. In addition, for some traditional enterprises, the change in culture and organizational structure is also an aspect that cannot be ignored in the implementation process, and there may be situations such as incompatibility with existing technologies. Figure 9 illustrates the relationship between food safety and big data, blockchain, and AI.

The application of artificial intelligence and big data in the food field can not only enhance production efficiency and product quality but also improve consumer experience and food safety. The emergence of emerging technologies derived from the combination of AI and big data, such as predictive analytics, automated production, intelligent quality inspection, personalized nutrition and food design, and intelligent marketing and personalized recommendation, will change the business model of the food industry, improve agricultural production efficiency, and provide a more traceable food supply chain. Future developments will need to focus on data privacy and security, talent development and technology diffusion, and environmental sustainability [163] to ensure the viability and sustainability of AI and big data in the food industry.

By leveraging the full potential of artificial intelligence and big data, the food industry can embrace a more efficient, safer, and sustainable development to provide better food products and services to consumers. This will play an important role in the future development of the global food industry.

## 5. Conclusions

This paper explores the application of big data and artificial intelligence (AI) in the food industry, critically analyzing the challenges these technologies face and the innovative solutions they offer. The research in this review shows that AI and big data have great potential and importance for the food industry. The application of these technologies has brought significant benefits to the food field, such as increased productivity, reduced costs, optimized supply chain management, and improved product quality and safety. However, we also note some of the challenges AI faces in the food industry, such as data privacy and security, technical complexity, and integration with traditional methods. With the continuous progress of technology, we can explore more advanced deep learning algorithms to improve the accuracy of detection in the food field, the combination of blockchain and big data as well as artificial intelligence to ensure the security of the data, and at the same time, the emergence of Industry 4.0 has accelerated the development of various technologies in the food industry, which is expected to realize a large number of industrial applications. In addition, the close integration of AI and big data will drive the food industry toward intelligence operations, sustainability, and innovation.

## Figures and Tables

**Figure 1 foods-12-04511-f001:**
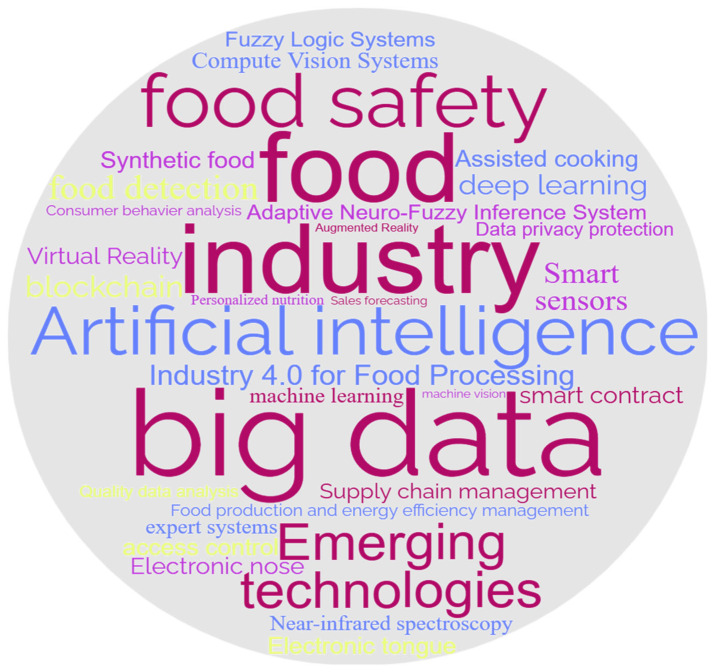
Word Cloud Generated Based on Keyword Retrieval in the Web of Science Database.

**Figure 2 foods-12-04511-f002:**
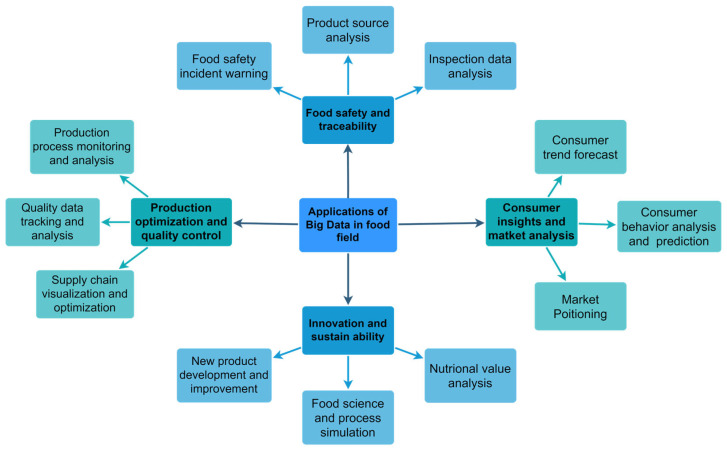
Applications of big data in the field of food safety and traceability, consumer insights and market analysis, production optimization and quality control, and innovation and sustainability.

**Figure 3 foods-12-04511-f003:**
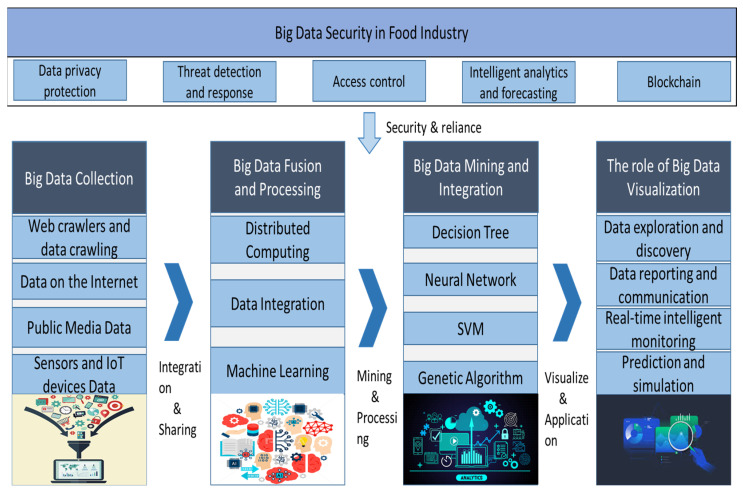
Five aspects of big data security in the food industry and the processing model of big data.

**Figure 4 foods-12-04511-f004:**
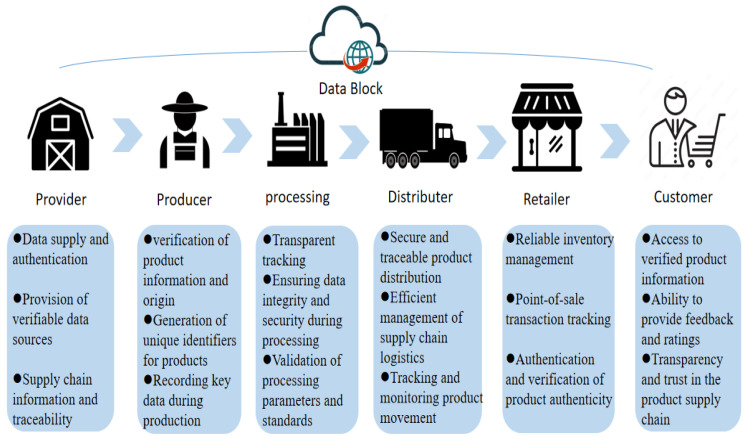
The process of the food supply chain in a blockchain system and the specific behaviors of different participants.

**Figure 5 foods-12-04511-f005:**
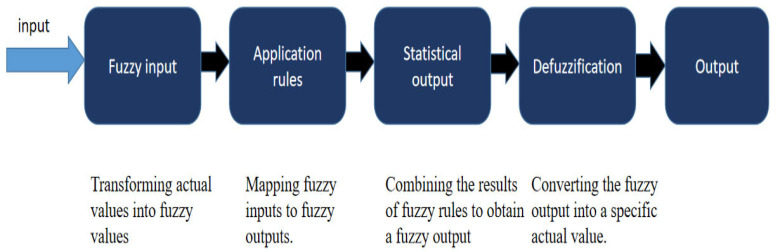
The operational steps of fuzzy logic controller.

**Figure 6 foods-12-04511-f006:**
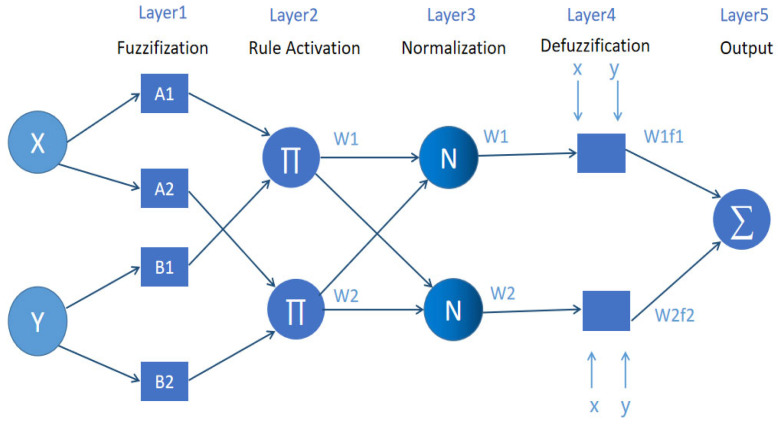
The five-layer structure of the ANFIS network.

**Figure 7 foods-12-04511-f007:**
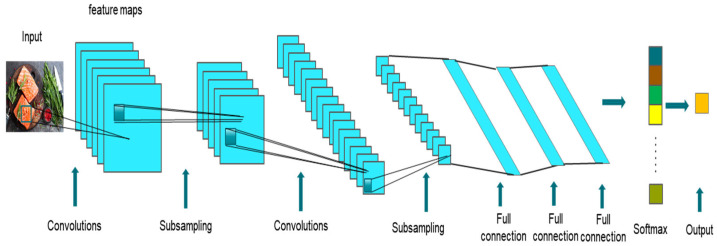
Automatic extraction of food shape features for food recognition using convolutional neural networks (CNN) in computer vision.

**Figure 8 foods-12-04511-f008:**
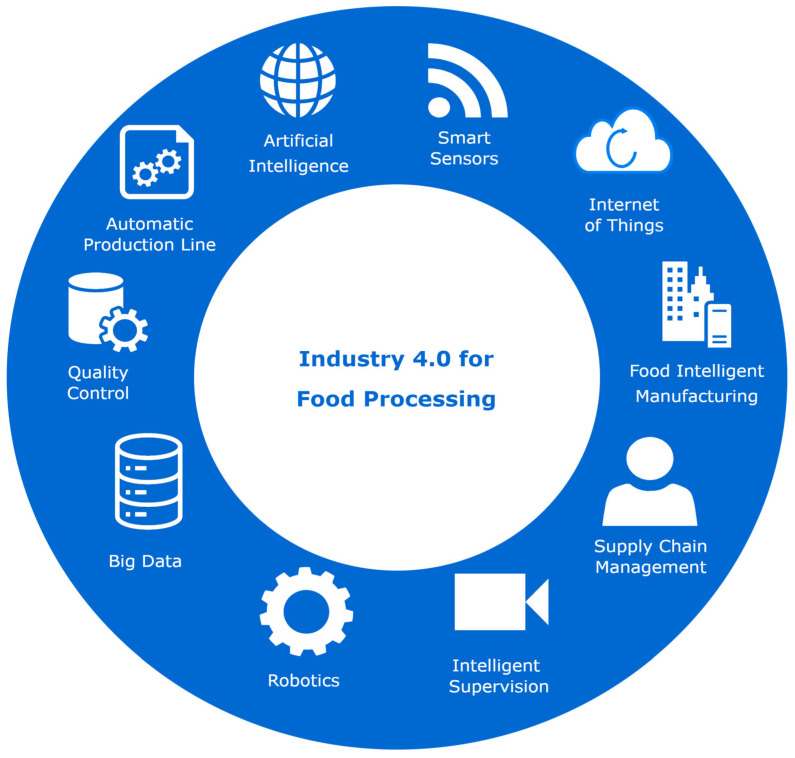
Key technology elements related to Industry 4.0 for food processing.

**Figure 9 foods-12-04511-f009:**
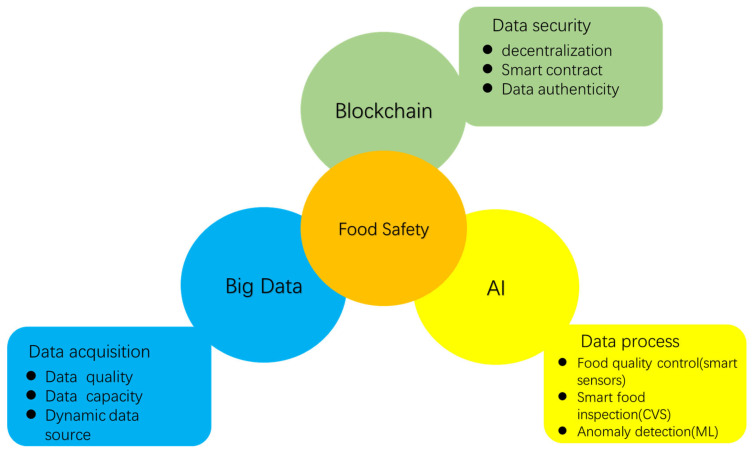
The relationship between food safety, big data, blockchain, and AI.

**Table 1 foods-12-04511-t001:** The application of blockchain in the supply chain of agricultural products.

Products	Objectives	Projects/Companies Involved
Beer	Tracking the entire production process of beer to reveal its relevant ingredients. (Downstream is the first company to apply blockchain technology to beer.)	Downstream Brewing Company [79]
Beef	Implement blockchain technology to detect its supply chain process and prevent food fraud.	BeefLedger Corporation [80]
Grain	Identify the entire supply chain.	Agri-Digital [81]
Mango	Guarantee the traceability of the mango production chain.	IBM, Wal-Mart, Nestle, etc. [82]
High fructose corn syrup	Supervision and management.	The Coca-Cola Company
Chicken	Ensure its traceability.	Gogochicken, OriginTrail Inc. [83]
Food waste	Monitoring and management, waste forecasting.	Plastic Bank, Agora Technology Labs
Rice	Supervision and to ensure the quality of rice during transportation.	“Agri-Food Blockchain” Project [13]
Milk	Traceability to prevent food fraud in the dairy production process.	“Agri-Food Blockchain” Project

**Table 2 foods-12-04511-t002:** Published ANFIS technology applications in the food industry.

Authors	Research Subjects	Expected Goals	Experimental Results
Arabameri et al. [112]	Olive Oil	Prediction of the quality of olive oil samples and determination of the influence of other factors	Highly accurate prediction of olive oil quality and successful prediction of the effects of time, temperature, and phenolics on its stability
Kaveh et al. [114]	Potatoes, garlic, and cantaloupe	Predicted moisture diffusion rate and energy consumption ratio	Successful use of the ANFIS model for accurate prediction of its water content
Mokarram et al. [115]	Orange	Predicting orange flavor	Successful use of the ANFIS model for accurate prediction of orange flavor
Abbaspour-Gilandeh et al. [113]	Quince	Prediction of kinetic energy and energy of quince under hot air drying	Accurate prediction of kinetic energy of quince using the ANFIS model and multiple linear regression
Kumar et al. [116]	Taro	Optimization of the extraction process of taro	Successful optimization of extraction process of taro bioactive compounds using response surface methodology and ANFIS
Ojediran JO et al. [117]	Yam	Predicting the drying characteristics of yam	Accurate prediction of drying characteristics of yam slices in convective hot air desiccant using ANFIS

**Table 3 foods-12-04511-t003:** Published computer vision applications in the food industry.

Authors	Research Subjects	Objectives	Experimental Results
Lopes et al. [125]	Barley flour	Forecast for barley flour	Classification using spatial pyramid segmentation method, the final prediction with SVM is 95%
Siswantoro et al. [129]	Eggs	Predicting egg volume	Successfully predicted egg volume with ANN model with a 97.38% success rate
Villager-Aguilar et al. [128]	Sweet pepper	Predicting the ripening status of bell peppers	Successfully developed an artificial vision system using CVS and ANN/FL to predict the ripeness of bell peppers with a maximum accuracy of 88% for FL and 100% for ANN

Bakhshipour et al. [130]	Iranian black tea and green tea	Classification of black and green teas in Iran	Successful classification of both with REP decision trees
Mazen et al. [131]	Banana	Predicting the ripening of bananas	Successfully used SVM and ANN algorithms to accurately predict the ripening level of bananas with an accuracy of 98%
Wan et al. [132]	Tomato	Predicting the ripeness of fresh tomatoes	Accurate detection of tomato ripeness with ANN algorithm with 99% accuracy
Markande et al. [133]	Potatoes	Grade classification of potatoes	A combination of CVS technology and fuzzy logic system successfully classifies potatoes and reduces costs
Garcia et al. [134]	Vegetable seeds	Sorting vegetable seeds	Successful classification of spinach seeds and cabbage seeds with ANN technology
Ozkan et al. [135]	Dry beans	Classification of different types of dry bean seeds	Successful classification of dry bean seeds with SVM, DT, ANN, and KNN algorithms
Zareiforoush et al. [136]	Rice	Grading the quality of rice	Successfully developed a system to grade rice quality with 97% accuracy

**Table 4 foods-12-04511-t004:** Comparison of conventional laboratory instruments, electronic nose, electronic tongue, computer vision, and sensory analysis.

Feature	Conventional Laboratory Instruments	Electronic Nose	Electronic Tongue	Computer Vision	Sensory Analysis
Fast detection	√	√	√	√	×
Low-cost analysis	√	√	√	√	×
Chemical free analysis	×	×	×	×	×
Objectivity	√	√	√	√	×
Non-destructive measurement	√	√	×	√	√
Sample pre-treatment	×	×	√	×	×
simple	√	√	√	√	×
Single operator	√	√	√	√	×
Permanent data storage	√	√	√	√	√

## Data Availability

Not applicable.

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
