# Peer review of "The Application of Artificial Intelligence and Big Data in the Food Industry"

_foods, 2023, doi:10.3390/foods12244511_

Round 1

Reviewer 1 Report

Comments and Suggestions for Authors

An interesting article, written in a way that encourages reading, and raises many important issues related to current trends in the food industry. One thing I missed was a critical look and indication of the risks and dangers associated with the implementation of new solutions and technologies. In addition, I would suggest correcting the graphics in the article, because in my opinion their quality and workmanship differ significantly from the level of the article. I recommend taking a closer look at the size of the drawings so that they fit in the page margins, maintaining the same style (e.g. if you start each description with a capital letter, continue this in the remaining descriptions, use a similar font size, Fig. 7 is proportionally much larger than the others and the font in it is also much larger than the one in the content of the article). The content of the drawings is interesting, but it is worth improving the way of presentation to make it more legible and, apart from the message it conveys, simply "pleasing to the reader's eyes". In the attached pdf file I have made some minor comments for the authors, please read them and comment or make corrections.

Reviewer 2 Report

Comments and Suggestions for Authors

The paper is an extensive exploration of the integration of Artificial Intelligence (AI) and big data within the food industry. It underscores the transformative impact of these technologies, discussing their applications in enhancing food safety, quality, production, supply chain management, and market insights. The review portrays a comprehensive landscape of the current state of AI and big data in the food sector, identifying their pivotal role in the industry's evolution and prospects for future development.

The discussion elucidates how AI technologies like machine learning, deep learning, and natural language processing, alongside big data analytics, are reshaping food production, processing, and retailing. It covers the historical context of the food industry, highlighting technological advancements and the establishment of comprehensive supply chain management systems. Furthermore, it delves into the significance of food safety, the emergence of digital transformation in the industry, and the challenges faced by these technologies.

Comments:

1) Could you elaborate on the specific instances where AI-driven technologies like machine learning or deep learning have demonstrated significant improvements in food safety or quality?

2) In what ways do the reviewed applications of AI and big data address the challenges associated with supply chain inefficiencies or waste reduction in the food industry?

3) Considering the complexity of data sources in the food sector, how were the privacy and security concerns surrounding big data addressed in the reviewed applications?

4) Were there instances identified where the integration of AI and big data resulted in tangible benefits for small-scale food producers or localized food systems?

5) Can you discuss the limitations or hurdles encountered in implementing AI and big data technologies within the food industry, particularly in terms of technical complexities or integration with existing systems?

6) Considering the prospects for future development outlined in the paper, what innovative approaches or technologies do you anticipate will further revolutionize the food industry by leveraging AI and big data?

Reviewer 3 Report

Comments and Suggestions for Authors

This article presents a critical analysis of the role and application of Big Data technologies and Artificial Intelligence (AI) in the food industry, in the context of Industry 4.0.

The article is very engaging and discusses the topic with the rigor that such a relevant topic requires. Moreover, the structure and the graphic material enhance readability and are well-designed. For all these reasons, I think that its publication in Foods is warranted.

Reviewer 4 Report

Comments and Suggestions for Authors

This manuscript submitted as a review does not have a methodology section describing how published articles were selected for analysis based on the journal guidelines for review papers. If this manuscript is to be presented as a review, I recommend that it be written again according to the journal guidelines for review papers. Also, more insight into domain knowledge in foods or agriculture is needed in article selection.

Round 2

Reviewer 2 Report

Comments and Suggestions for Authors

 Accept in present form.

Author Response

Dear Reviewer:

Thank you for your valuable appreciation and approval of our review paper. We appreciate your kind comments very much. Thank you for taking the time to review our manuscript. We are honoured to have the opportunity to receive feedback from someone with your level of expertise.

Best wishes,

Haohan Ding
                                                             on behalf of
                                                             all authors

Reviewer 4 Report

Comments and Suggestions for Authors

On p.2, I would like to thank the authors for adding a paragraph of their journal selection of review papers. From their updated description, I realized that the authors reviewed "review papers" not original articles. I would like to ask the authors to clearly explain in the manuscript why review papers were used for this review manuscript instead of original articles.

In the updated paragraph on p.2, the authors claimed that high-frequency words are visually highlighted to help readers quickly perceive important information. From Figure 1, the term "blockchain" is a visually highlighted word even bigger than food safety, which is beyond my expectations. A review article should present an unbiased summary of the current understanding of the topic or field. As a peer viewer, I must assess the selection of studies that are cited in the paper. So I searched the Web of Science to see if blockchain is mentioned frequently in food-related review papers. From the 1,297 review papers, I found no review paper that has blockchain in the title (see attached table). I believe blockchain technology is one of the essential fields in the food industry. However, I could not find a single review paper with blockchain in the title when I tried to follow what the authors described. I recommend that this manuscript be written again to describe in detail how the studies that were cited by the manuscript were selected.

Comments on the Quality of English Language

p.2 line 90, "To ensure access to the latest research results" - it looks like an incomplete sentence.
